# Chemo-Thermo-Mechanical FEA as a Support Tool for Damage Diagnostic of a Cracked Concrete Arch Dam: A Case Study

**Noemi Schclar Leitão [1],\*** and **Eloísa Castilho [2]**

1　Laboratório Nacional de Engenharia Civil (LNEC), Av. do Brasil 101, 1700-066 Lisbon, Portugal

2　Instituto Superior Técnico, Universidade de Lisboa, Av. Rovisco Pais 1, 1049-001 Lisbon, Portugal; eloisa.castilho@tecnico.ulisboa.pt

\*　Correspondence: nschclar@lnec.pt

**Abstract:** Most of the larger hydropower plants in Western Europe, the former Soviet Union, North America and Japan were constructed between the 1940s and 1970s. This implies that the rehabilitation or repair of existing dams is a top priority, which entails new challenges for the dam engineering community. Since no two dams are the same, in cases in which abnormal behavior is suspected, an in-depth diagnosis of the state of the dam to define the causes and consequences of the damage is required. To illustrate the diagnostic process, an old concrete arch dam is presented which showed signs of reservoir water seepage through some construction joints, resulting in a buildup of calcium carbonate on the downstream face. After analyzing the available data, we put forward a hypothesis that the high temperature gradient promoted the opening of some construction joints on the upstream face during the first filling of the reservoir. Over time, water penetration expanded the cracks, reaching the downstream face. To prove our diagnosis, a chemo-thermo-mechanical finite element analysis was carried out in order to simulate the behavior of the dam during its construction and initial impoundment.

**Keywords:** concrete arch dam; finite element chemo-thermal-mechanical analysis; diagnosis; first filling





## 1. Introduction

Between the 1940s and 1970s, spurred initially by World War II and followed by strong post-war economic and population growth, state-owned utilities built significant hydropower developments throughout Western Europe, as well as the former Soviet Union, North America and Japan [1]. These water infrastructural projects were critical to economic development of the agriculture and energy sectors. Since the 1980s, however, the number of new dams in developed countries has started to decline. This slowdown in dam construction resulted from several political-ecological factors: (i) the economics of construction (the best and most economical sites were exploited first); and (ii) the increased awareness of the need to consider the environmental impact of large dams [2,3].

Concurrently, water availability and climate change have been topics of increasing concern in recent decades. To face this situation without constructing new dams, developed countries have no other option than continue using the existing dams as much as possible [4]. This implies that the repair, rehabilitation or strengthening of existing dams should be carried out in order to enhance their performance, extend their service life and increase their load-carrying capacity.

Similar to other concrete structures, concrete dams are inherently durable and usually require a minimum amount of repair and maintenance. However, with the passage of time, the exposure of a dam to various external and internal aggressions can lead to the deterioration of the structure.

One of the most harmful distress mechanisms affecting the durability and serviceability of aging structures is the alkali–aggregate reaction (AAR). Due to its impact on the

safety and durability of concrete structures, many efforts to harmonize and coordinate the diagnosis, prognosis and assessment of evaluating AAR damage are underway through the International Union of Laboratories and Experts in Construction Materials, Systems and Structures (RILEM) [5,6].

However, there are many causes of concrete damage in dams apart from AARs [7,8]. Freezing–thawing and drying–wetting cycles, structural overload, cracking due to seismic actions or non-uniform foundation movements, thermal and shrinkage volumetric changes, cavitation, abrasion–erosion, and sulfate attack are some examples of causes of damage to concrete. Design and construction defects, poor-quality concrete, poor finishing and poor curing can lead to concrete suffering damage.

In addition, there are a number of other factors that may affect service life, resulting in the necessity of strengthening dams. Changes in the design criteria (hydrology and seismic hazards) based on new information obtained since the initial design of a dam, changes in methods of analysis, and new safety concepts or the results of risk assessments (new risks and changes in risk acceptance criteria) can trigger the need to strengthen actions [8].

The planning, design, implementation and monitoring of a repair and/or strengthening project should always begin with a careful assessment of the existing structure. The purpose of this assessment is to identify all defects and damage, to diagnose their causes and hence to assess the present and likely future adequacy of the structure. The information obtained from the structural assessment can then be used to determine whether corrective work is required. Without prior planning and proper assessment, any program of corrective work is likely to prove ineffective [9].

However, historically, the repair of concrete dams has been based at least as much on art as on science [7]. Determining the cause of damage has often been given very little importance by technicians, who make decisions based on their expertise and intuition [10].

In order to improve practices and knowledge in concrete dam repair, the U.S. Bureau of Reclamation, which operates and maintains hydroelectric and water resource structures in the Western United States, has invested a lot of effort into formulating a consistent and systematic approach to repairing concrete. In this regard, the importance of a correct diagnosis was emphasized:

> The first and very important step of repairing damaged or deteriorated concrete is to correctly determine the cause of damage. Knowing what caused the damage, and reducing or eliminating that cause, will make the repair last longer. If no attempt is made to eliminate the original cause of damage, the repair may fail as the original concrete did, resulting in wasted effort and money (von Fay [7], p. 1–13).

With the purpose of organizing a diagnostic procedure, Pardo-Bosch and Aguado [10] and Blanco et al. [11] outlined a framework to aid in the diagnosis of different pathologies that affect concrete dams based on the theory used in medical diagnosis. In this context, numerical models have been proven to be important tools for the study and validation of hypotheses elaborated during this diagnosis procedure [12–14].

Following the above approach closely, the present article illustrates the diagnostic process of an old concrete arch dam, which shows signs of reservoir water seepage through some construction joints, resulting in a buildup of calcium carbonate on the downstream face. After analyzing the available data, we put forward a hypothesis that the high temperature gradient promoted the opening of some construction joints on the upstream face during the first filling of the reservoir. Over time, water penetration expanded the cracks, reaching the downstream face. To prove this diagnosis, a chemo-thermo-mechanical finite element analysis (FEA) was carried out in order to simulate the behavior of the dam during its construction and initial impoundment.

## 2. Diagnosis Procedure

According to [10,11], the sequence of activities can be divided into two main stages:

1. The first stage, which usually lasts a few weeks, involves the following:
   a. Clinical history;
   b. Filed (dam) inspection;
   c. Initial cabinet works;
   d. First hypothesis.
2. The second stage, which may take weeks to months, entails the following:
   a. Laboratory tests;
   b. Numerical modeling;
   c. Validation of the hypothesis;
   d. Prediction of future behavior.

## 3. Dam Description

The dam studied in this article was completed in 1955 and is located in the central region of Portugal. It is a double-curvature, thin concrete arch dam with 17 keyed monoliths, a maximum height of 63 m and a crest length of 175 m at an elevation of 181 m. The width is 2 m at the crest and 7 m at the base of the largest monolith. The dam has a surface uncontrolled spillway along its crest designed for a maximum discharge capacity of 2200 m$^3$/s. The reservoir's normal level is 175 m. A concrete pad (called socle in Portugal) was added to the foundation to make the site symmetrical and to provide a better distribution of stresses on the foundation, as shown in Figure 1.

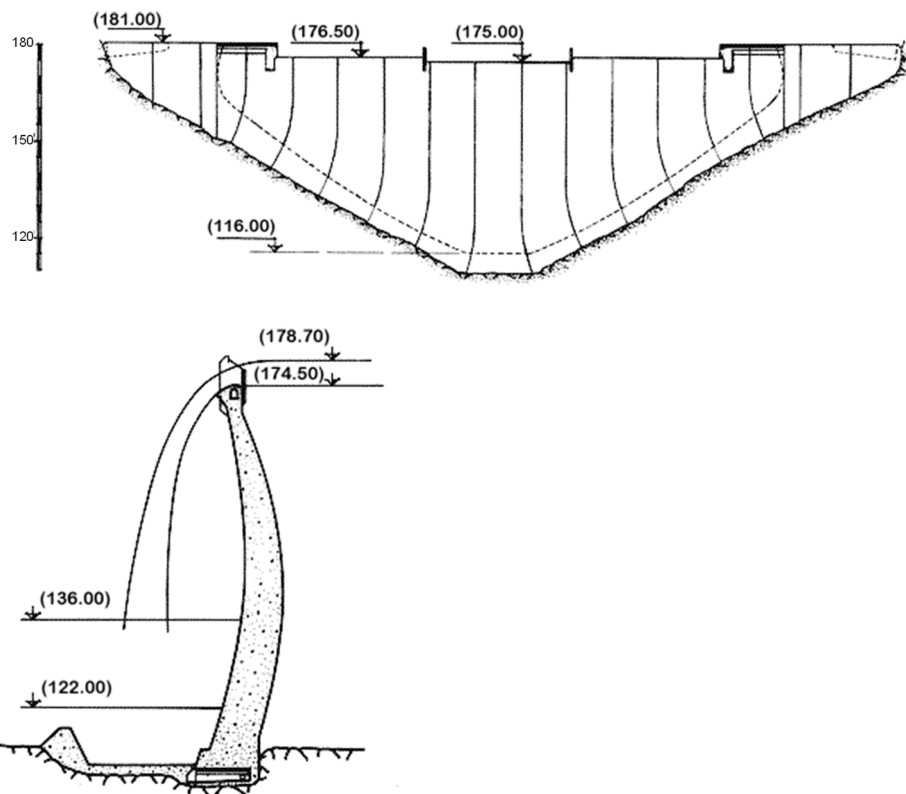

**Figure 1.** Upstream view and vertical cross section of the dam.

The dam foundation corresponds to a contact zone of Cambrian and Silurian formations, with the special feature being that the left bank consists of granite and the right bank consists of schist. Both rocks are very siliceous and, in general, the rock mass is fractured and presents weathering near the surface [15].

The monitoring system consists of several devices to measure concrete and air temperatures, water level, displacements in the dam and its foundation, joint movements, strains and stresses in the concrete, pressure and discharges in the foundation.

Because of the lack of pendulum, the geodetic surveying method is the exclusive source of horizontal displacements. This planimetric system consists of triangulation networks on the downstream banks, as shown in Figure 2.

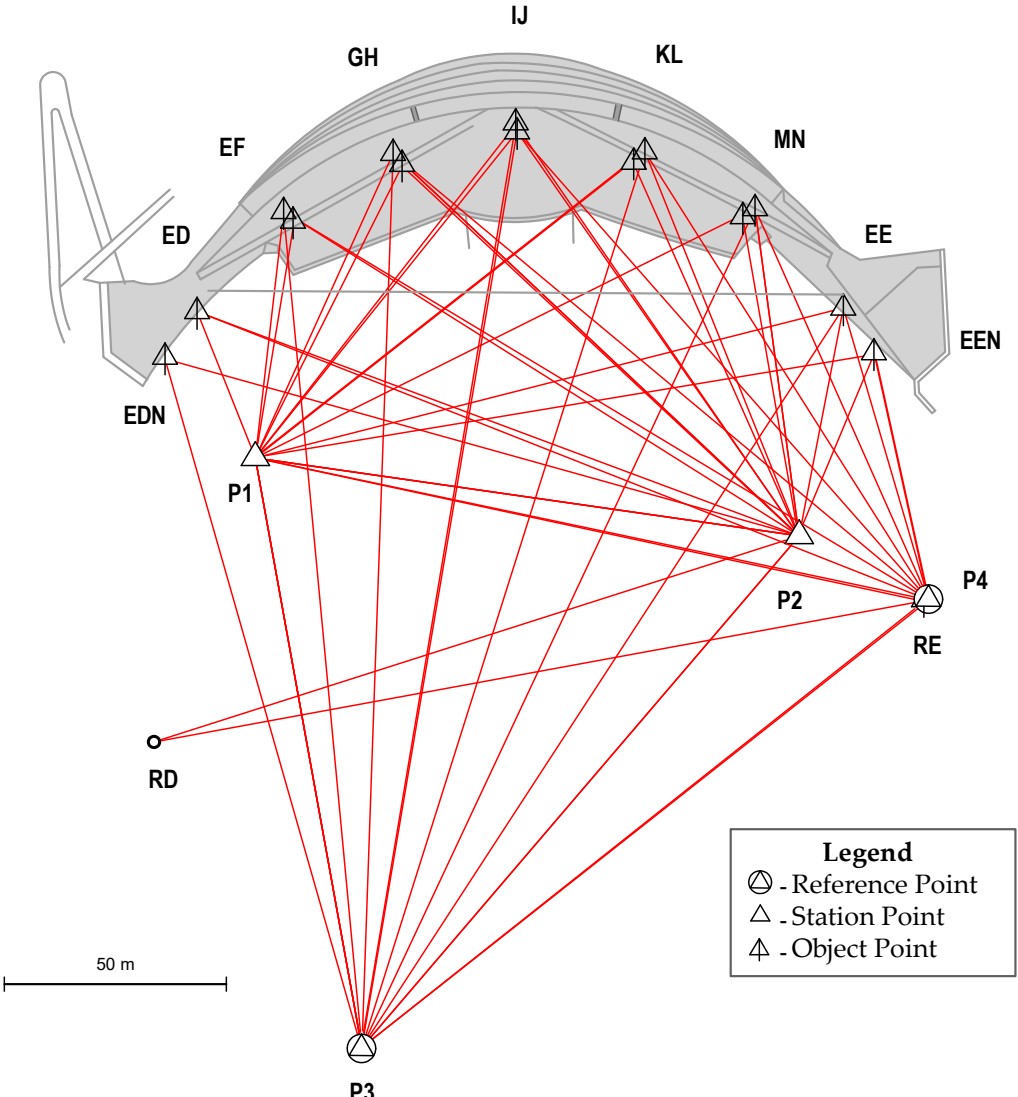

**Figure 2.** Triangulation network.

## 4. Clinical History and Field Inspections

Apart from the already cited article [15] published by the former owner of the dam, Hidroeléctrica do Zêzere, which describes the preliminary studies, design, construction and material tests of the dam, most of the information used in the present work comes from internal technical reports by the National Laboratory for Civil Engineering (LNEC). However, it is important to note that the most relevant results of these reports have also been previously published in different conferences and journals [16–19]. Description of the concrete composition used in the dam and other relevant thermal information are also available in Silveira's thesis [20].

### 4.1. Dam Design

It is important to note that in the 1950s, two different approaches were used to perform stress analyses of dams on both sides of the Atlantic. In the USA, the design of dams was based on an extensive analytical method, the so-called trial-load method, developed by the U.S. Bureau of Reclamation. Meanwhile, European countries, mainly France, Italy and

Portugal, preferred the use of small-scale structural models rather than the time-consuming trial-load method to find the final shape of the dam. As a result, the arch dams constructed in Europe were thinner than those constructed in the USA [21,22].

In the present case, the preliminary design of the dam was based on the "independent arch" theory. According to this theory, the dam was assumed to be divided by horizontal planes into arch rings of units of vertical height. These rings were considered to work as independent arches.

For the final design of the dam, small physical (or scale) model tests were carried out in the LNEC. The tests were performed in homogeneous plaster–diatomite models at a scale of 1/200. Figure 3 illustrates the principal stress path obtained using brittle varnish technique, the average principal stresses obtained in the tests for a reservoir water level of 177.5 m and the composition of these stresses with the stresses computed by analytical method for the self-weight of the dam.

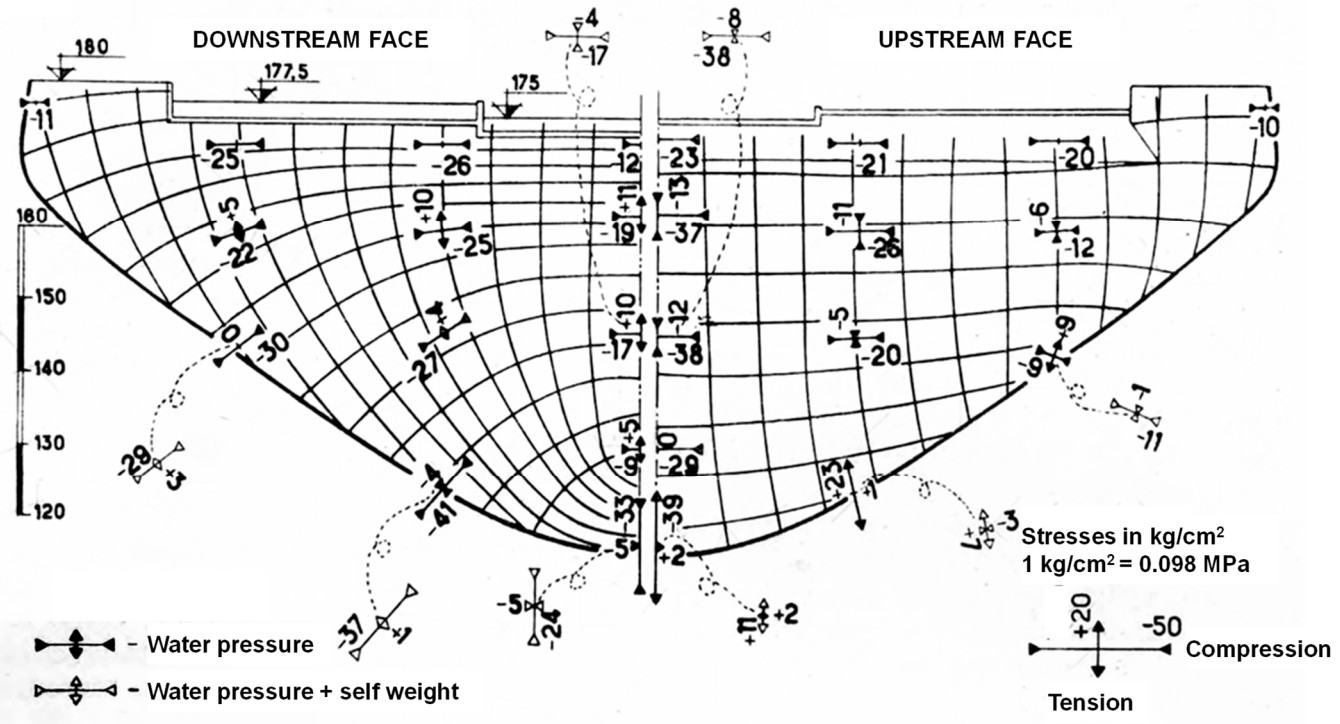

**Figure 3.** Principal stress on the downstream face (on the left) and on the upstream face (on the right) obtained by model tests (adapted from [16]).

At the same time, in order to reach a more meaningful estimate of the margin of safety, tests up to failure were also performed. To this aim, a technique was employed of loading with jacks that simultaneously increased the dead weight and water load, thus reproducing progressive reductions in the concrete strength, as shown in Figure 4.

*4.2. Dam Construction and Initial Impoundment*

The construction of the dam took place between October 1954 and September 1955. The first filling of the reservoir began in 1955, with the dam still under construction.

The grouting of the contraction joints was carried out in various stages between May 1955 and March 1956. During May and June 1955, with the dam still under construction, the grouting of the lower part was performed, between the foundation and the level at 135 m. The middle part between the levels at 135 m and 155 m was performed in August 1955. Finally, the grouting of the upper part took place between February and March 1956, in correspondence with the first emptying of the reservoir.

Between August and September 1958, a second emptying of the reservoir took place, however no work was reported at that time.

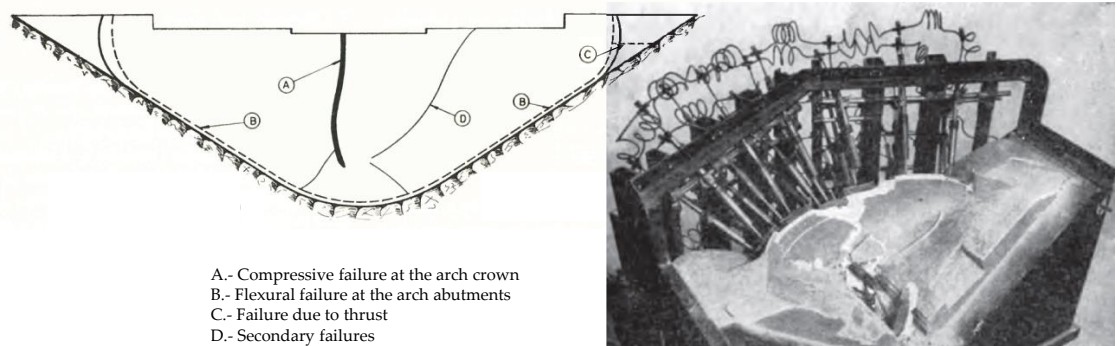

A.- Compressive failure at the arch crown
B.- Flexural failure at the arch abutments
C.- Failure due to thrust
D.- Secondary failures

**Figure 4.** Tests up to failure of the dam (adapted from [18,23]).

### 4.3. Cracking Evolution

The existence of water seepage through some construction joints has been known since 1976. As time passes, the leaching from the concrete has been forming calcium carbonate deposits on the downstream face of the dam. Besides these water leaking cracks, several construction joint openings also exist at a lower level.

Figure 5 shows the evolution of the downstream face's appearance over nearly 25 years. The oldest photograph of the damage that the authors could find goes back to some time before 1992 and corresponds to the inventory of dams in Portugal published in 1992 [24]. The rest of the photographs correspond to the periodic inspections carried out by LNEC since 2002. The major change observed in this period corresponds to January 2007, when a new seepage path through another construction joint appeared in the monolith G–H.

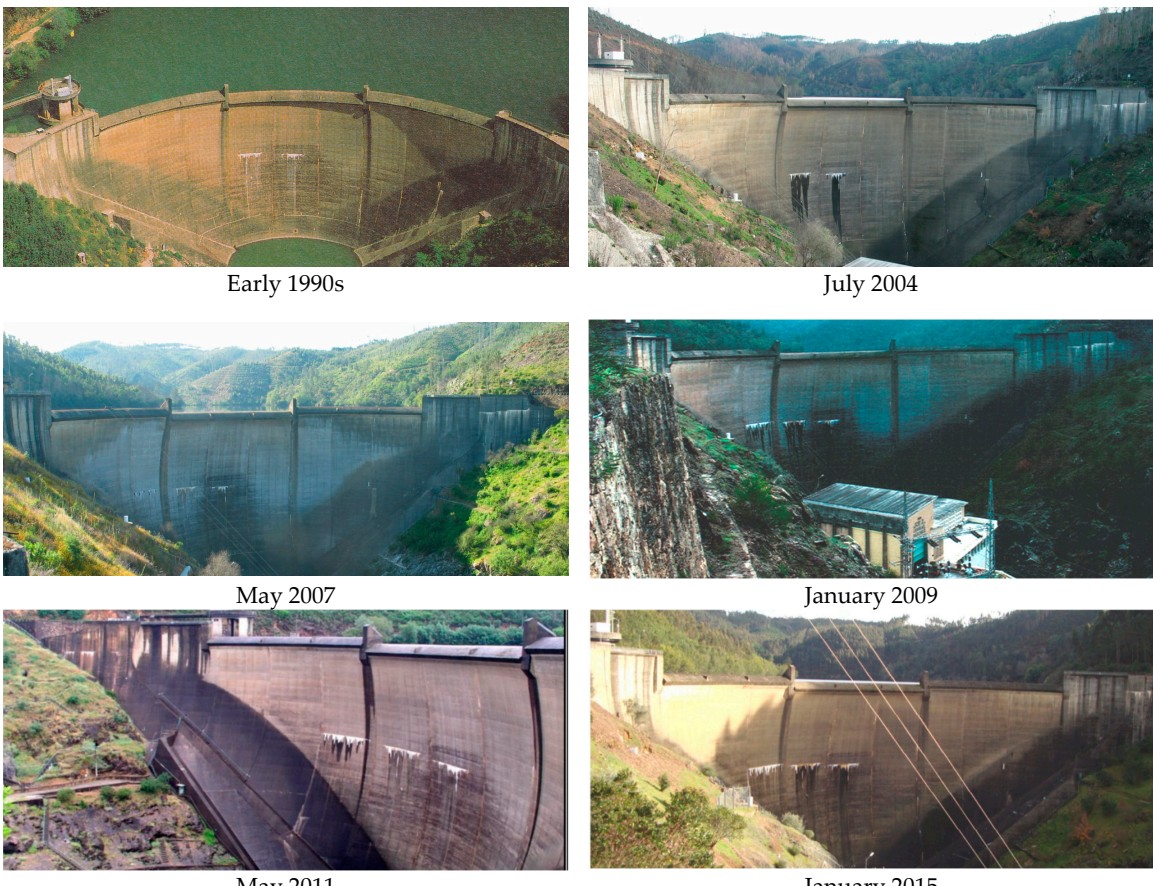

| Early 1990s | July 2004 |
| May 2007 | January 2009 |
| May 2011 | January 2015 |

**Figure 5.** Downstream surface appearance from early 1990s to January 2015.

As part of the safety inspection, the owner of the dam has performed periodical crack surveys since 1982. These surveys allow us to identify and locate the existent cracks, as well as follow their evolution, as shown in Figure 6.

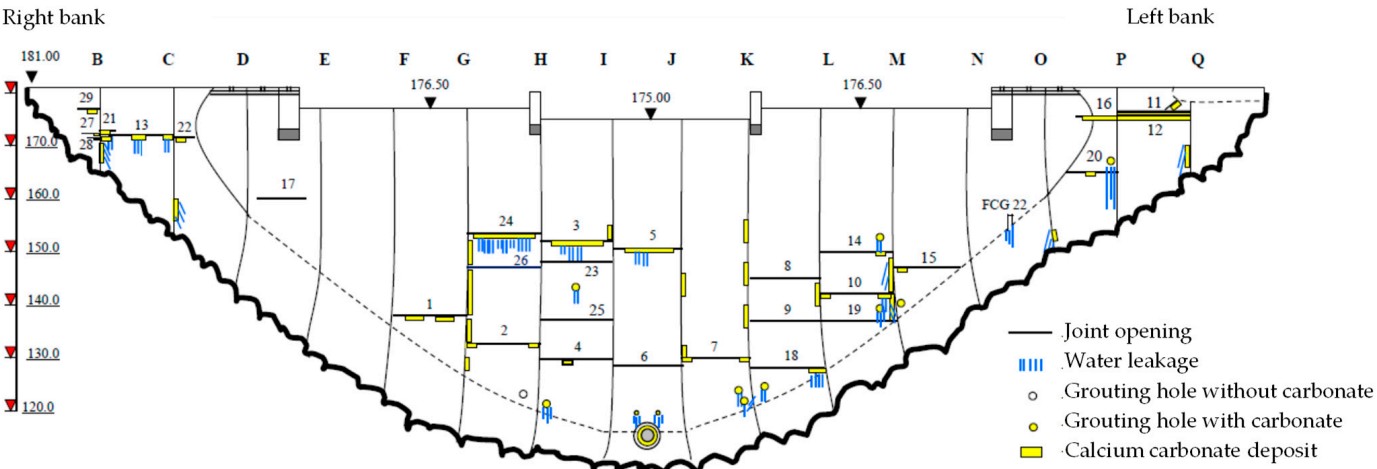

**Figure 6.** Crack survey carried out in March 2012.

In addition, some cracks were instrumented with joint meters. Figure 7 shows, from top to bottom, in the first graph, the monthly average temperature and the water level, and in the following three graphs, the monitored opening of joint meters installed in crack 3, crack 5 and crack 7, respectively, for the period between 2002 and 2016.

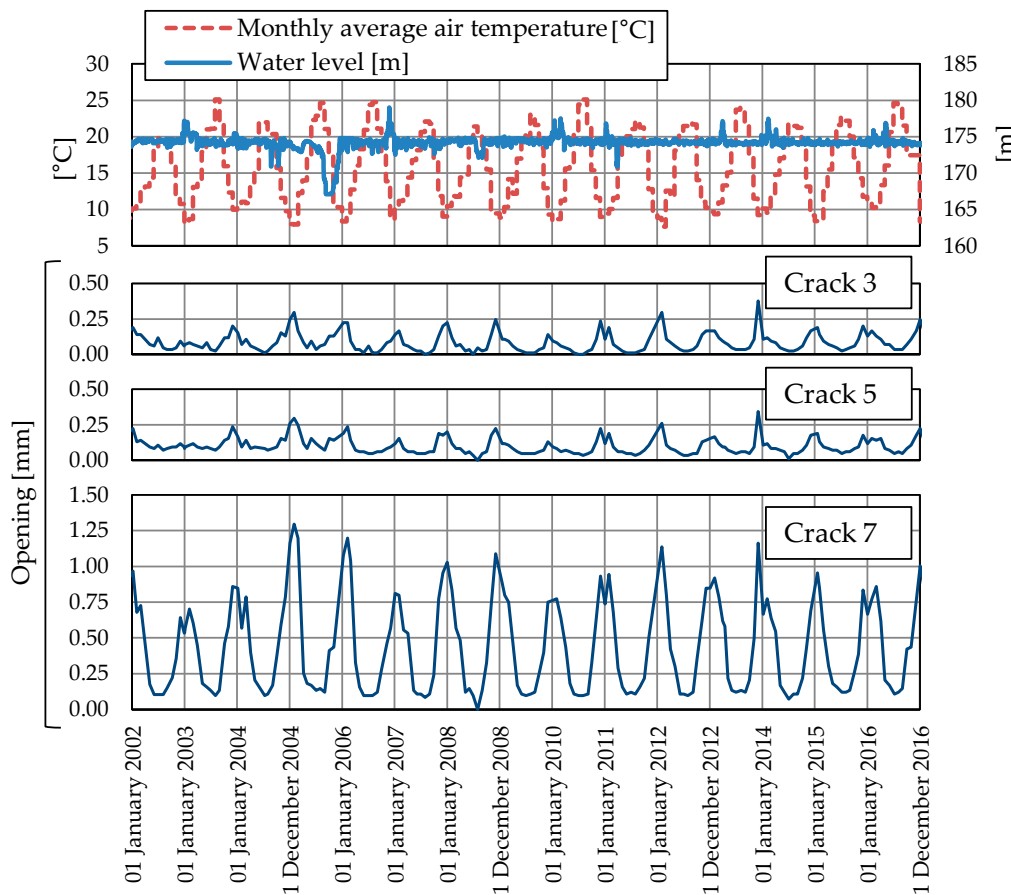

**Figure 7.** Monthly average air temperature, reservoir levels, and monitored crack openings between 2002 and 2016.

## 5. Cabinet Works and Initial Diagnosis Hypothesis

Apart from the open construction joints, the dam did not present any other significant signs of concrete deterioration. The classic symptom of map cracking induced by AAR was not observed. Nevertheless, the displacements of the dam were analyzed in order to detect any signs of irreversible displacements. As it has already been established, in arch dams, internal concrete swelling phenomena usually induce upstream drift and crest rises, even before cracking takes place. Since irreversible displacements were not detected, any swelling processes could be excluded from our study.

Additionally, with respect to progressive deformation, it is important to note that, although there is no sign of non-recoverable deformation in the no-stress strain meters located in most of the dam, the no-stress strain meters installed near the foundation have been showing signs of concrete swelling since 2001. However, this phenomenon is limited and does not justify, at this moment, further investigation.

After excluding an ongoing swelling process effect, thermal stresses were identified as the main cause of the opening of the construction joints.

It is worth noting that the properties of construction joints are greatly influenced by how the joints are prepared before pouring the next lift of concrete. Even with good preparation, the strength and fracture energy at a construction joint are much lower than the values for mass concrete, creating horizontal planes of weaknesses [25]. Therefore, it is not surprising that vertical tensile stresses were released by the opening of the nearest construction joints.

As the open construction joints with water seepage (cracks 3, 5 and 24) show a very slight opening seasonal variation compared with the cracks situated under the level of 150 m, as illustrated by Figure 7, the phenomenon associated with each type of cracking was considered as a different type.

For the lower cracks, the high seasonal fluctuation together with the absence of seepage suggest that they are thermal fatigue cracks due to the seasonal thermal variation acting on the downstream face of the dam. These fatigue cracks do not propagate a lot in depth. Thus, they do not compromise the behavior of the structure under static loads and their formation will be not studied in this paper.

On the other hand, the formation of the cracks located over the level of 150 m was associated with the fact that the initial impoundment started with the dam under construction. The upper concrete lifts did not have enough time to release all the heat of hydration before being in contact with the cold water. Once opened, the water pressure within the cracks triggered additional damage and weakened the fracture properties. This effect coupled with the hydrostatic pressure slowly opened a water path to the downstream face, leaching calcium carbonate.

In the following section, the formation of the cracks over the level at 150 m due to thermal stresses generated during the initial impoundment will be investigated using FEA.

## 6. Finite Element Model

In their guidelines for nonlinear FEA of existing concrete structures and infrastructures, Hendrix et al. remark:

> A finite element model of a structure is an abstraction of the physical structure with a number of assumptions, generalizations, and idealizations. The abstraction process has two distinct steps: first, the abstraction from the structure to the mechanical model, and then the abstraction from the mechanical model to the finite element model. In the first step, assumptions and simplifications have to be made regarding to which extent and to which detail the structure has to be modeled, how the boundaries of the model are described, which loads on the structure are significant and how they are described, et cetera. The second step is to discretize the mechanical model into a finite element model, and attach the necessary attributes such as material models, boundary conditions, and loading to the finite element model (Hendrix et al. [26], p. 9).

Indeed, FEA requires a great number of a priori modeling decisions. These decisions require a certain level of expertise and are quite subjective in nature, leading to considerable differences in the approaches adopted. As pointed out by Saouma and Hariri-Ardebili [27] (p. 243), "the selected finite element analysis is often a compromise between: (a) needs and time constraint, (b) our understanding of the problem and of nonlinear analysis, (c) tools available, and (d) quality of results expected".

Based on previous experience in modeling the behavior of concrete arch dams during initial impounding [28], a chemo-thermal-mechanical model is elaborated to investigate the crack formation during the first filling of the reservoir. First, a chemo-thermal analysis using the chemical affinity concept is carried out to determine the temperature distribution during the construction and the initial impoundment of the dam. Then, a nonlinear viscoelastic analysis is performed in order to obtain the structural response of the dam, with particular focus on the nonlinear behavior caused by the opening and closure of the contraction joints as well as the interface between the dam and the foundation.

### 6.1. Finite Element Mesh

For the finite element analysis, the double curvature concrete arch dam and an adequate volume of the foundation are represented as shown in Figure 8.

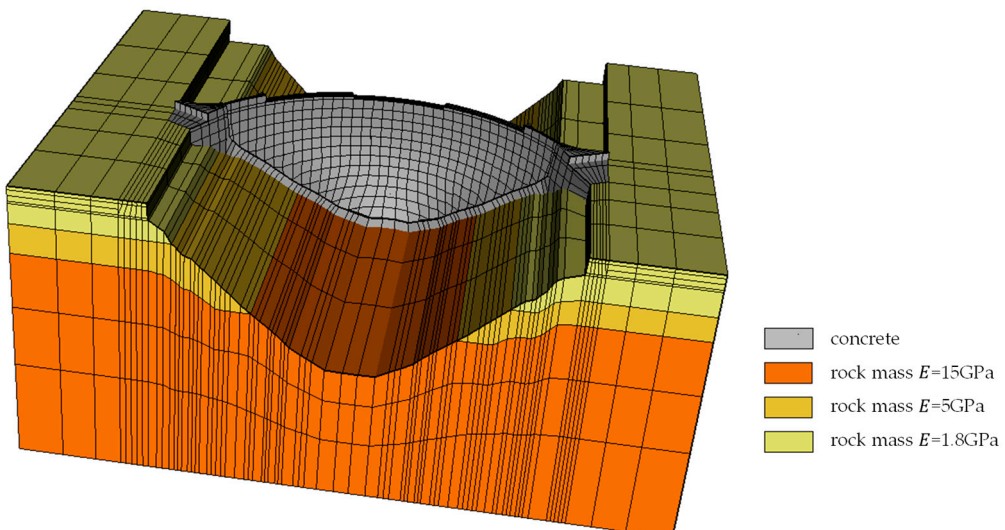

**Figure 8.** Finite element model of the dam.

The dam model comprises four layers of 20-node solid elements. To facilitate the transmission of data from the chemo-thermal analysis to the mechanical analysis, the same solid elements' mesh is utilized for both the thermal and the mechanical analysis.

In order to take into account the influence of the contraction joints in the mechanical analysis, the arch is divided into 17 monoliths. The contraction joints and the dam–foundation interface are represented by 16-node zero-thickness interface elements. The model consists of 5200 solid elements and 968 interface elements.

### 6.2. Thermal Analysis

6.2.1. Governing Equations

For a stationary medium, the transient heat conduction equation is given as follows:

$$\frac{\partial}{\partial x}\left[k_x \frac{\partial T}{\partial x}\right] + \frac{\partial}{\partial y}\left[k_y \frac{\partial T}{\partial y}\right] + \frac{\partial}{\partial z}\left[k_z \frac{\partial T}{\partial z}\right] + G = \rho\, c\, \frac{\partial T}{\partial t} \tag{1}$$

with the following boundary conditions:

$$T = \overline{T} \quad \text{in } \Gamma_T \tag{2}$$

$$k_x \frac{\partial T}{\partial x} l + k_y \frac{\partial T}{\partial y} m + k_z \frac{\partial T}{\partial z} n + q_c + q_r + q_q = 0 \quad \text{in } \Gamma_q \tag{3}$$

and the following initial condition:

$$T = T_o \quad \text{in } \Omega \text{ for } t = t_o \tag{4}$$

where $t$ is the time; $T$ is the temperature; $k_x$, $k_y$ and $k_z$ are the thermal conductivities; $G$ is the internally generated heat per unit of volume and time; $\rho$ is the material density; $c$ is the specific heat; $\bar{T}$ is the temperature at the boundary $\Gamma_T$; $q_c$ is the heat flux due to convection; $q_r$ is the heat flux due to radiation and $q_q$ is the solar radiation absorption at the boundary $\Gamma_q$; $l$, $m$ and $n$ are the direction cosines; and $T_o$ is the temperature at time $t_o$. It is noteworthy that the convention in expression (3) is positive when heat flux flows outwards from the body [29].

The convective heat transfer between the structure's surface and the air is influenced by the wind speed and air temperature. The heat gain or loss from a surface due to convection is given as follows:

$$q_c = h_c(T - T_a) \tag{5}$$

where $h_c$ is the convection heat transfer coefficient and $T_a$ is the air temperature.

Due to the difference in temperature between the structure's surface and the surrounding air, the surface of the structure emits electromagnetic radiation known as thermal radiation. This radiation is measured by the Stefan–Boltzmann law as follows:

$$q_r = \varepsilon \, \sigma \left( T^4 - T_a^4 \right) \tag{6}$$

where $\varepsilon$ is the emissivity of the surface and $\sigma$ is the Stefan–Boltzmann constant given as $5.669 \times 10^{-8}$ W(m$^2$ K). However, when $T$ and $T_a$ are close, which is the normal condition in civil engineering structures, it is possible to rewrite (6) in a quasi-linear form as follows:

$$q_r = h_r(T - T_a) \tag{7}$$

where $h_r$ is the radiation linear coefficient defined as follows:

$$h_r = \varepsilon \, \sigma \left( T^2 - T_a^2 \right) (T - T_a) \tag{8}$$

As a result, the total heat transfer can be calculated by combining the contribution of both heat transfer mechanisms, convection and radiation, leading to the definition of a new coefficient called total thermal transmission coefficient, $h_t$. In essence, this new coefficient is a convection heat transfer coefficient that is updated to consider radiation.

Finally, the solar radiation boundary condition is given as follows:

$$q_q = a \, I_T \tag{9}$$

where $a$ is the absorption coefficient and $I_T$ is the solar irradiance.

### 6.2.2. Chemo-Thermal Model

Different mathematical formulations as a function of time, representing the evolution of heat of hydration in adiabatic conditions, have been proposed. In reality, however, hydration does not evolve adiabatically, and the heat source is greatly affected by the actual values of the temperatures that develop inside the concrete. As a result, the time parameter by itself is insufficient to accurately describe the progress of the hydration reaction. Due to the reciprocal relationship between the rate of hydration and the temperature of the concrete, it is necessary to explicitly model the effect of the actual temperature and the

rate of reaction when calculating both the temperature development and the hardening process [30].

The evolution of the hydration reaction is represented by an Arrhenius-type equation that takes into consideration the thermo-activation and exothermic nature of the reaction, following the hydration kinetic model based on the thermodynamics of multiphase porous media proposed by Ulm and Coussy [31], in which the variation of the skeleton mass (reaction velocity) $\frac{dm}{dt}$ in [mol/s] is expressed as follows:

$$\frac{dm}{dt} = \frac{1}{\eta} A \exp\left(\frac{E_a}{RT}\right) \tag{10}$$

where $\eta$ is a viscosity term representing the increase in physical barrier of calcium silicate hydrates (CSH), which tends to isolate the cement grain from the free water, and depends on the state of the hydration reaction; $A$ is the affinity of the chemical reaction or, in other words, the thermodynamic force associated to the rate of hydration formation, which also depends on the state of the hydration reaction; $E_a$ is the apparent thermal activation energy, which is considered to be constant with relation to the hydration degree; $R$ is the universal constant of gases [8.314 J/(mol K)]; and $T$ is the temperature in K.

For practical reasons, it is helpful to rewrite the model in terms of the normalized variable called hydration degree, defined as the relation between the mass of the skeleton at time $t$ normalized by the mass of the skeleton when hydration is complete, i.e., $\xi(t) = m(t)/m_\infty$

$$\dot{\xi} = \widetilde{A}(\xi) \exp\left(-\frac{E_a}{RT}\right) \tag{11}$$

where $\dot{\xi}$ is the time derivative of $\xi$, and the function $\widetilde{A}(\xi)$ is the normalized affinity which completely characterizes the macroscopic hydration kinetics for a given concrete mixture.

Then, the problem of heat transfer during concrete hydration is obtained from Equation (1), substituting the internally generated heat per unit of volume and time $G$ by the term $L\dot{\xi}$, where $L$ represents the latent heat of hydration of the material:

$$\frac{\partial}{\partial x}\left[k_x \frac{\partial T}{\partial x}\right] + \frac{\partial}{\partial y}\left[k_y \frac{\partial T}{\partial y}\right] + \frac{\partial}{\partial z}\left[k_z \frac{\partial T}{\partial z}\right] + L\dot{\xi} = \rho\, c\, \frac{\partial T}{\partial t} \tag{12}$$

The simultaneous solution of the two last equation represents the thermochemical coupling, which is a nonlinear problem in $T$ and $\xi$.

Among the different empirical relationships used to represent the normalized affinity $\widetilde{A}(\xi)$, the empirical relationship reported by Cervera et al. [32] was adopted in this work:

$$\widetilde{A}(\xi) = \frac{k_\xi}{\eta_{\xi o}}\left(\frac{A_{\xi o}}{k_\xi \xi_\infty} + \xi\right)(\xi_\infty - \xi)\exp\left(-\bar{\eta}\frac{\xi}{\xi_\infty}\right) \tag{13}$$

where $\xi_\infty$ is the asymptotic degree of hydration, and $k_\xi$, $A_{\xi o}$, $\eta_{\xi o}$ and $\bar{\eta}$ are material properties.

### 6.2.3. Thermal Properties

The thermal properties of the concrete were determined using the method reported by the U.S. Bureau of Reclamation in [33]. This method is based on the mix proportions and petrographic composition of aggregates. It assumes that each material composing the concrete contributes to the conductivity and specific heat in proportion to the amount of the material present in the concrete. Taking into account that the average composition of the concrete used in the dam, expressed in part by weight, was 1:1.66:6.66 (cement:sand:granite) with a water/cement ratio of 0.52 [20], and assuming a reference temperature of 20 °C, we found that $k = 2.65$ W/(m °C) and $c = 866$ J/(kg °C).

Table 1 lists the concrete and rock mass foundation parameters that were employed in the thermal analysis.

**Table 1.** Thermal material properties.

| Properties | Rock Mass Foundation | Concrete |
|---|---|---|
| Density $\rho$ [kg/m$^3$] | 2657 | 2460 |
| Specific heat $c$ [J/(kg °C)] | 715 | 866 |
| Thermal conductivity $k$ [W/(m °C)]] | 4.91 | 2.65 |

The computation of the S-shape function representing the heat of hydration of the concrete was based on the data given in [15], which refers to three samples of cement tested at the ages of 3, 7 and 28 days. Additionally, the heat at the ages of 90 and 365 days was estimated applying a percentage of increase based on the 28 days' value given in [20], resulting in the following:

$$Q = A \exp\left(\frac{B}{t}\right) \tag{14}$$

where $Q$ is the heat of hydration in [kJ/m$^3$], $t$ is the time in [h], $A$ = kJ/m$^3$ and $B$ = −65 h, as shown in Figure 9.

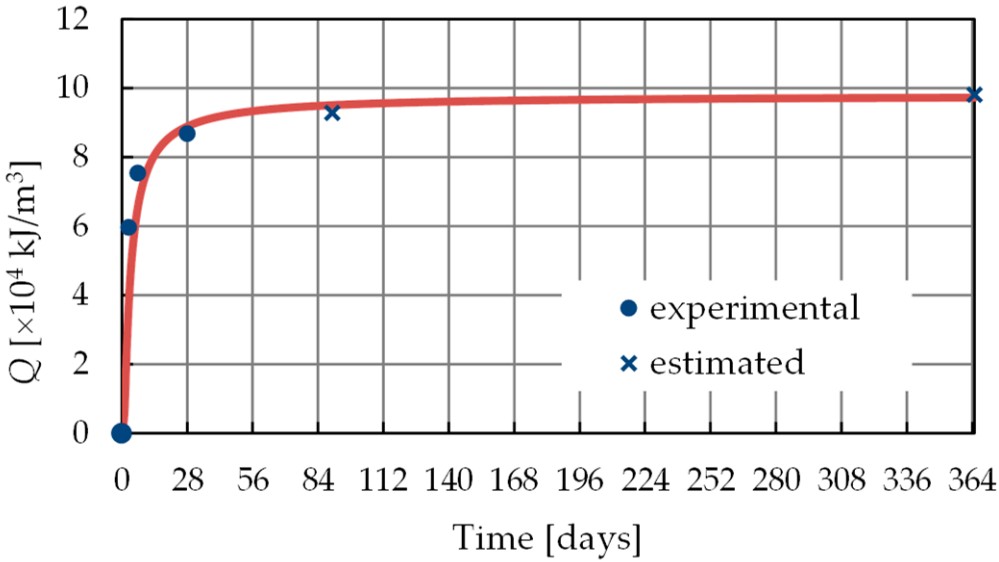

**Figure 9.** Concrete heat of hydration development.

After applying the approach presented in [34] and assuming $\xi_\infty = 0.75$ and $E_a/R = 4000$ K, the parameters of Equation (13) result in the following: $k_\xi/\eta_{\xi o} = 34.17$ s$^{-1}$; $A_{\xi o}/k_\xi = 1.34 \times 10^{-2}$; and $\bar{\eta} = 6.26$.

### 6.2.4. Boundary Conditions

Convection/radiation heat transfer and solar radiation flux absorption boundary conditions were used for the dam's air-exposed boundaries. Fixed reservoir water temperature boundary conditions were used for the dam's submerged boundaries. Convection/radiation heat transfer boundary conditions were used for the air-exposed boundaries. Furthermore, adiabatic boundary conditions at the lateral faces and a fixed temperature boundary condition of 12 °C at the bottom were applied to the artificial boundaries of the rock mass foundation.

The average monthly temperature recorded at the dam site during the construction, initial impoundment and first period of operation, along with a daily variation extrapolated from more recent readings, were used to estimate the daily air temperature variation. The

solar irradiance was represented by an exponential function derived from the graphics presented in [20] for the different Portuguese regions:

$$\frac{I_b}{\cos Z} = I_o \exp(-0.968 + 0.760 \cos Z) \tag{15}$$

where $I_b$ is the beam component of the solar radiation, $Z$ is the solar zenith angle and $I_o$ is the solar constant (1367 W/m$^2$).

The total heat transfer coefficient $h_t$ was set to a constant value of 24 W/(m$^2$ K) for the whole model, with the exception of the formwork-insulated surfaces. Due to the lack of information about the formwork characteristics, an empirical value of 0.10 times the total heat transfer coefficient of the concrete was adopted, yielding $h_t$= 2.4 W/(m$^2$ K). The concrete absorption coefficient was assumed as 0.5.

As mentioned before, the reservoir water temperature was introduced as a prescribed temperature. For the initial period, from the start of the impoundment to 31 December 1956, a constant value over the completed depth of 11 °C was adopted. After 31 December 1956, the approximation given by Bofang [35] was adopted as follows:

$$T^{water}(y, d) = T_m^{water}(y) - T_a^{water}(y) \cos\left\{\frac{2\pi}{365}[d - d_o(y)]\right\} \ [°C] \tag{16}$$

with

$$T_m^{water}(y) = 11 + 5 \exp(-0.16y) \ [°C] \tag{17}$$

$$T_a^{water}(y) = -6 \exp(-0.067y) \ [°C] \tag{18}$$

$$d_o(y) = 4450[1 - \exp(-0.00038y)] + 24.6 \ [\text{days}] \tag{19}$$

where $y$ is the depth of the water; $d$ is the fractional day of the year; and $T_m^{water}$, $T_a^{water}$ and $d_o$ are the annual mean temperature, the amplitude of annual variation and the phase difference of water temperature at depth $y$, respectively.

More details about how to apply dam boundary conditions can be seen in [36].

### 6.2.5. Concrete Placement Schedule

Due to the lack of sufficient data on the concrete placement schedule, it was determined using the data available in the original LNEC technical reports by trial-and-error processes. In particular, the quarterly construction progress (Figure 10a), the volume of concrete pouring per month (Figure 10b) and the installation date for embedded monitoring devices were used.

The concrete placement of the dam was performed with a lift thickness of 2 m and an interval of placement of 3 days. However, the lift thickness and the intervals of placement adopted in the simulation have to be adapted to the 5 m height of the finite element used to represent the dam, resulting in one-week interval of placement. Therefore, the "birth" of a new element in height occurred, at least, one week after the "birth" of the underlying element. Figure 10c shows the final one-week interval placement schedule adopted for the simulation.

### 6.2.6. Analysis and Results

The in-house code PATQ [34], which uses a fully implicit backward Euler finite difference scheme for the time discretization and a finite element scheme for the spatial discretization, was used to perform the transient thermal analysis. Due to the dependence of the hydration rate on temperature, it uses a two-level iterative procedure. At the structural level, the iteration is caused by the nonlinear dependence of the "thermal body force" on the temperature. At the local level (that is, at each integration point), the iteration is a

result of the nonlinear dependence of the degree of hydration (internal variable) on the temperature (free variable).

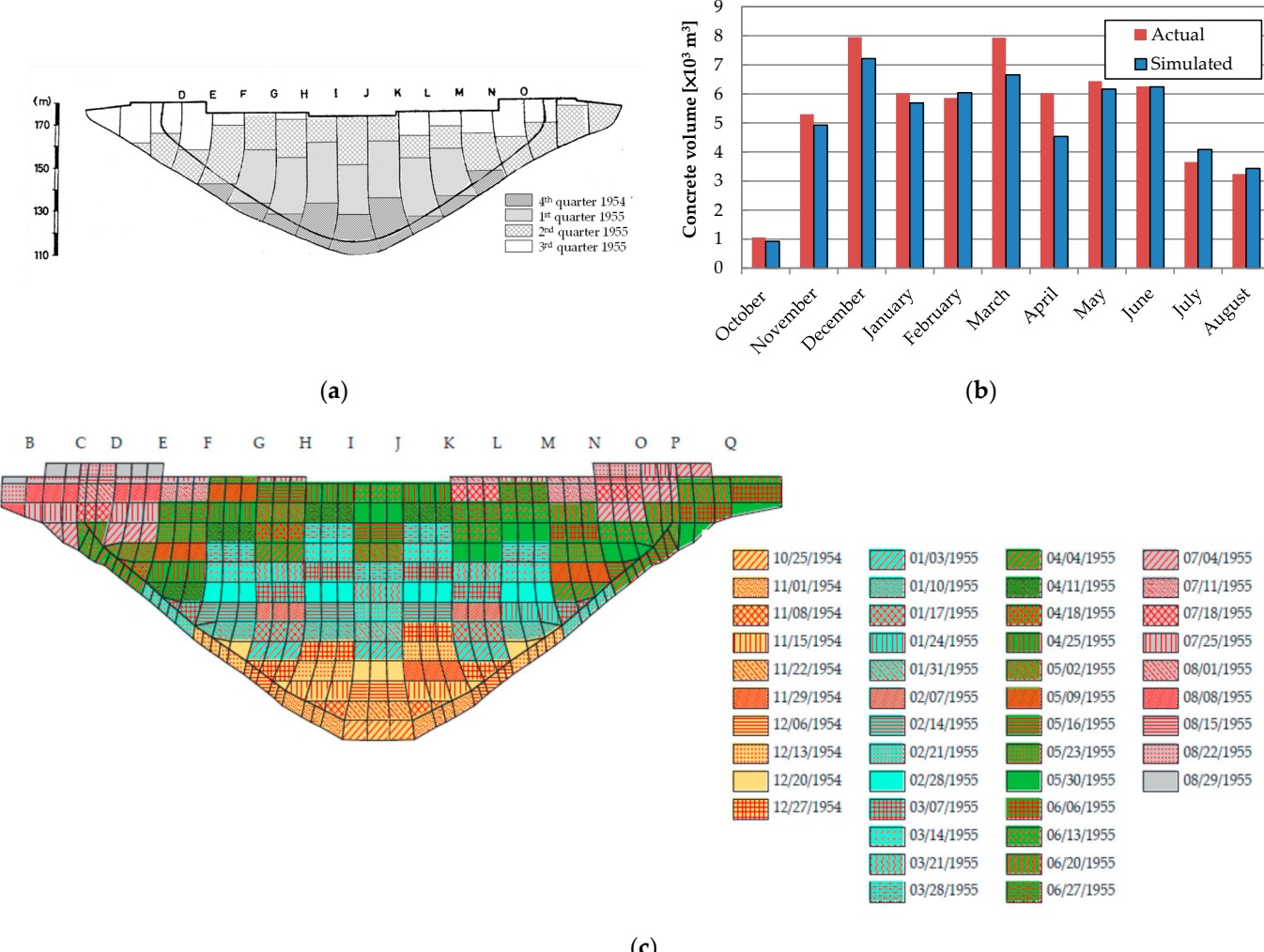

**Figure 10.** Concrete placement schedule: (**a**) quarterly construction progress; (**b**) comparison of the actual and simulated volume of concrete pouring per month; (**c**) concrete placement schedule adopted for the simulation.

The birth and death approach was used to update the finite element model at each step of construction. The analysis was carried out using an incremental time stepping of 1 h.

To evaluate the performance of the model, the simulation was performed over an extended period from the beginning of the construction, in October 1954, to December 1958. In this way, the simulation encompasses the construction phase, the first filling of the reservoir and the two first times the reservoir was emptied.

A comparison of the predicted and measured temperatures was performed to validate the model. Figure 11 compares the temperatures measured by thermometers T12, T14, T17, T20 and T22 located at an elevation of 150 m in monolith I–J to those obtained with the numerical model at the same position.

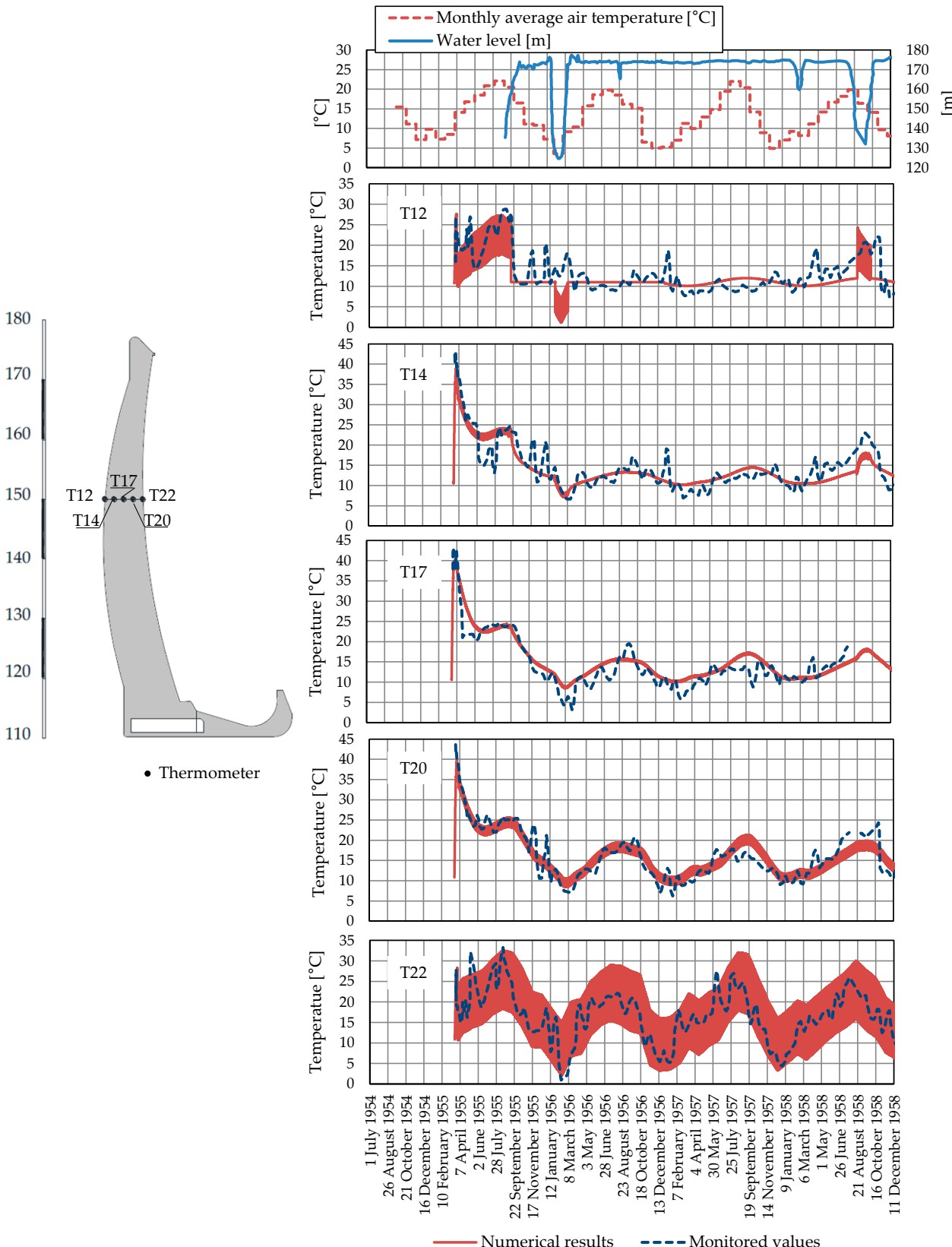

**Figure 11.** Monthly average air temperature, reservoir levels and comparison of predicted and monitored temperatures at thermometers in monoliths I–J at an elevation of 150 m.

It is important to note that monitoring data measured before 1970 were stored in the form of hand-drawn lines charts. Therefore, WebPlotDigitalizer [37] was used in order to extract the corresponding numerical data for the analysis.

Overall, between the numerically calculated temperatures and the monitored data, a very good agreement was observed. Consequently, we had confidence that the thermal model could accurately describe the temperature loads.

### 6.3. Mechanical Analysis

Lombardi's paper, which was presented at the Dam Fracture Workshop in Locarno in September 1990, pointed out that a dam:

> ... is not a simple, ideal, elastic body. Vertical contraction joints subdivide the dam in blocks. Even after grouting they will represent a discontinuity in as much as they cannot be grouted to the dam faces but only to the water-stops. Additionally, the grouting is seldom carried out in a completely satisfactory manner. Furthermore, the concrete is placed in layers and lifts making weakness planes possible at frequent elevations, and inhomogeneities and anisotropies – which can hardly be detected – are likely to exist in the concrete mass (Lombardi [38], p. 4).

Therefore, as the above quotation suggests, a model considering elastic, isotropic, and homogeneous materials which are free from discontinuities is not a true representation of the concrete of the dam or the rock mass foundation. Nevertheless, it will be always necessary to adopt some degree of simplification. For one, it is impossible to know and more so to represent all the inhomogeneities, anisotropies and discontinuities present in the dam and its foundation. Moreover, an idealized simple model can better explain and characterize the dominant features of the physical phenomena studied.

Therefore, in order to characterize the actual behavior of the dam, two different finite element models were adopted. The two models were selected in order to demarcate the bandwidth within which the behavior of the dam is expected. The first one is a linear continuous model, which is the standard model used in a dam analysis. In the second model, the nonlinear model, the contraction joints and the dam–foundation interface are represented explicitly by no-tension interface elements.

### 6.3.1. Mechanical Properties

In order to take into account the creep effect, the double power law [39] was adopted for the concrete as follows:

$$J(t - t') = \frac{1}{E_o} + \frac{\varphi_1}{E_o} \left( t'^{-m} + \alpha \right) \left( t - t' \right)^n \tag{20}$$

where $J(t - t')$ is the compliance function (or the creep function), i.e., the strain at age $t$ caused by a unit of uniaxial constant stress acting since age $t'$; $E_o$ is the asymptotic modulus; and $n, m, \alpha$ and $\varphi_1$ are the material parameters.

Actually, as the finite element code uses the library MATPAR given by Bažant in [40], the input data for expression (20) are expressed in terms of the following five parameters: $E_{28}$, $E_o/E_{28}$, $n, m$ and $\alpha$. By calibrating the model with the recently recorded displacements, $E_o$ = 40.5 GPa was obtained. Based on this value and using the standard relationship $E_o/E_{28} = 1.5$, $E_{28}$ was fixed in 27 GPa. The remaining three parameters $n, m$ and $\alpha$ were estimated using the empirical formulas given in [39] for $w/c$ = 0.52 and $f'_c$ = 27.5 MPa.

The rock mass foundation was considered as a linear elastic material. According to the rock field tests carried out by LNEC, the rock mass foundation was divided in three different zones, reflecting that near-surface rock is more weathered and fractured, as shown in Figure 8.

Zero-thickness interface elements were used to represent the contraction joints and the dam–foundation interface. In this formulation, the contact constraint is enforced by the

penalty method, where the normal stiffness $k_n$ and the tangential stiffnesses $k_s$ and $k_t$ play the role of penalty coefficients. This means that they have to be set as high as possible to guarantee that no penetration takes place while the joints are closed, but not so high to avoid ill-posed problems. Hence, the normal and tangential stiffness have no physical meaning, eliminating the need for extra experimental investigations. Moreover, the assumptions of no-tension and no-sliding conditions to characterize the normal and tangential behavior, respectively, also help us avoid the need for additional material parameters.

The material parameters utilized in the mechanical analysis are listed in Table 2.

**Table 2.** Mechanical material properties.

| Material | Properties | Values |
|---|---|---|
| Concrete | Double power law | |
| | $E_0$ [GPa] | 40.50 |
| | $n$ | 0.12 |
| | $m$ | 0.34 |
| | $\alpha$ | 0.048 |
| | $\varphi_1$ | 1.78 |
| | Poisson's ratio $\nu$ | 0.20 |
| | Coefficient of thermal expansion $\alpha$ [1/°C] | $10^{-5}$ |
| Rock mass foundation | Young's modulus $E$ [GPa] | 15.00, 5.00 or 1.80 |
| | Poisson's ratio $\nu$ | 0.20 |
| | Coefficient of thermal expansion $\alpha$ [1/°C] | 0.00 |
| Joints | $k_s = k_t$ [GPa/m] | 2000.00 |
| | $k_n$ [GPa/m] | 2000.00 |
| | $f_t$ | 0.00 |

### 6.3.2. Loads

The analysis was carried out considering the dead load, normal water load and internal strains caused by temperature changes.

The dead load corresponds to the weight of the concrete, and it was applied staggered at the "birth" of each element. As the contraction joints are open during the construction phase, the corresponding interface elements were considered inactive at this stage in order to simulate the cantilever behavior of the monoliths during construction.

The normal water load corresponds to the hydrostatic pressures acting on the dam's upstream face resulting from the reservoir. For the computation of the normal water load, the pressure is considered to vary linearly with depth and to act normally on the dam surface.

The thermal load was computed from the temperature variation obtained in the chemo-thermal analysis.

The uplift load at the dam–foundation interface was ignored because the dam is very thin.

### 6.3.3. Analysis and Results

The phases of the construction, the initial impoundment and the first period of operation were solved incrementally through time. The in-house code PAVK [28] was used for this analysis.

During the construction phase, the time intervals were constrained by the concrete placement and formwork striking dates. After that, until 30 April 1956, a smaller time interval of 1 or 2 days was used in order to follow the effect of the reservoir rise on the upstream face of the dam. Finally, the analysis was completed with a two-week interval, except when a geodetic survey was carried out.

Figure 12 compares the radial displacements obtained with the numerical model with the corresponding values measured by geodetic survey triangulation. This figure shows, from top to bottom, in the first graph, the monthly average temperature and the rising reservoir water level; and in the following three graphs, the comparison of the radial displacements at elevations of 170 m, 150 m and 130 m in block I–J. The negative direction indicates that the dam moved downstream.

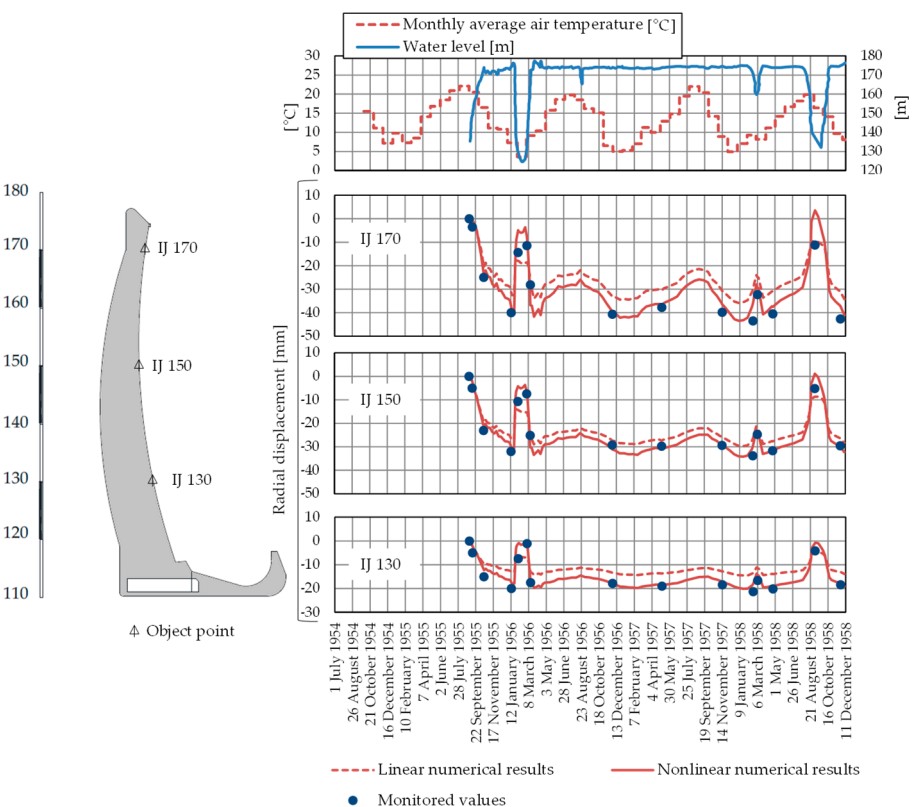

**Figure 12.** Monthly average air temperature, reservoir levels and comparison of predicted and monitored radial displacements in monoliths I–J at elevations of 170 m, 150 m and 130 m.

As can be observed, the nonlinear model shows a better agreement with the monitored radial displacements than the linear one. The comparison of the models shows that during impoundment and emptying periods, the nonlinear model reflects larger displacements than the linear one. This apparent greater flexibility of the nonlinear model is due to the opening/closing of the contraction joints and the dam–foundation interface in response to the stress changes.

In contrast, when the reservoir is full, leading to the closing of the contraction joints, both models show almost similar displacement fluctuations. This fact corroborates that the greater flexibility in the response of the dam for periods of low reservoir levels is due to the movements of the discontinuities and not to a lower concrete modulus of elasticity.

The importance of considering the discontinuities in the model is also confirmed by Figure 13, which shows the vertical stresses estimated and measured near the upstream heel and the downstream toe of monoliths I–J.

The measured stresses were recorded by stress meters located at 1 m from the faces of the dam. It is worth noting that the stress meter is a device that allows the direct measurement of compressive stress. In contrast with strain meters, the stress meter is fully responsive to compressive stress and very closely indicates the true stress at all times without any further analysis, i.e., no elasticity modulus needs to be adopted, and without regard to deformation due to causes other than stress [41].

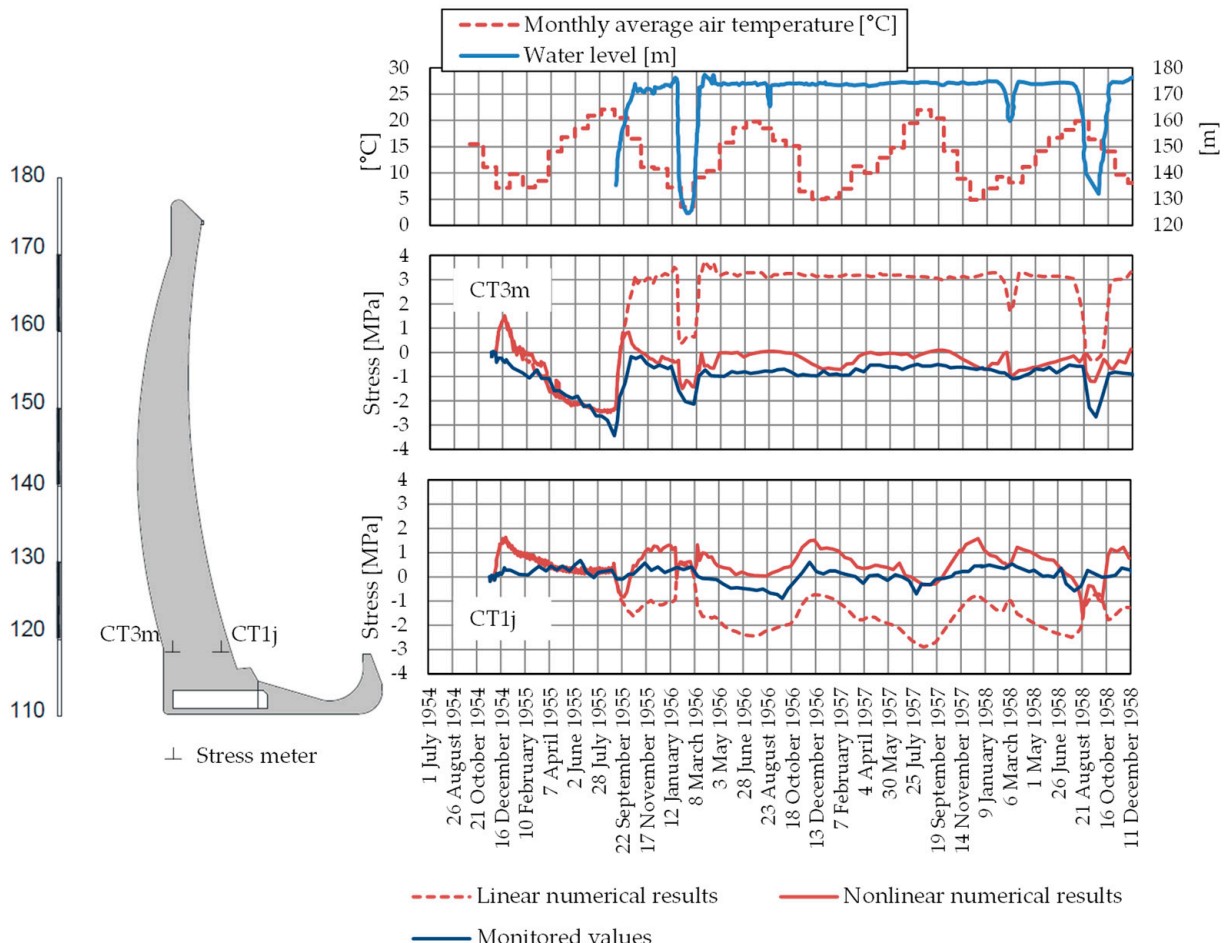

**Figure 13.** Monthly average air temperature, reservoir levels and comparison of predicted and monitored vertical stresses in monoliths I–J at elevations of 118 m.

Analyzing Figure 13 (stress meter CT3m), it can be observed that when the reservoir reaches approximately the level of 160 m, the two models no longer stay together. This is because tensile stresses appear in the upstream heel of the dam, inducing the opening of the corresponding dam–foundation interface element. This behavior is fully validated by the stresses measured in stress meter CT3m.

The opening of the interface element also modifies the stresses at the downstream toe of the dam, as shown by Figure 13 (stress meter CT1j). This is due to the redistribution of the stresses, which causes the decrease in the cantilever stresses at the expense of an increase of the arch stresses. This fact is validated by the stress meter CT1j.

## 7. Validation of the Diagnosis Hypothesis

According to the previous analysis, the nonlinear model is the most representative model of the behavior of the dam. Therefore, this model was used to study the cracking of the upstream face of the dam.

Figures 14 and 15 show the principal stress plots obtained during the initial impoundment over the upstream face of the dam. The surface stresses were computed in the center of each external face by extrapolating the stresses obtained in the integration points.

The length and direction of the arrows show the relative magnitude and direction of compression (blue) and tension (red).

The initial impoundment started on 3 September 1955 with an empty reservoir. The rise in the reservoir level took place at a very high rate, reaching a level of 160 m on 26 September 1955.

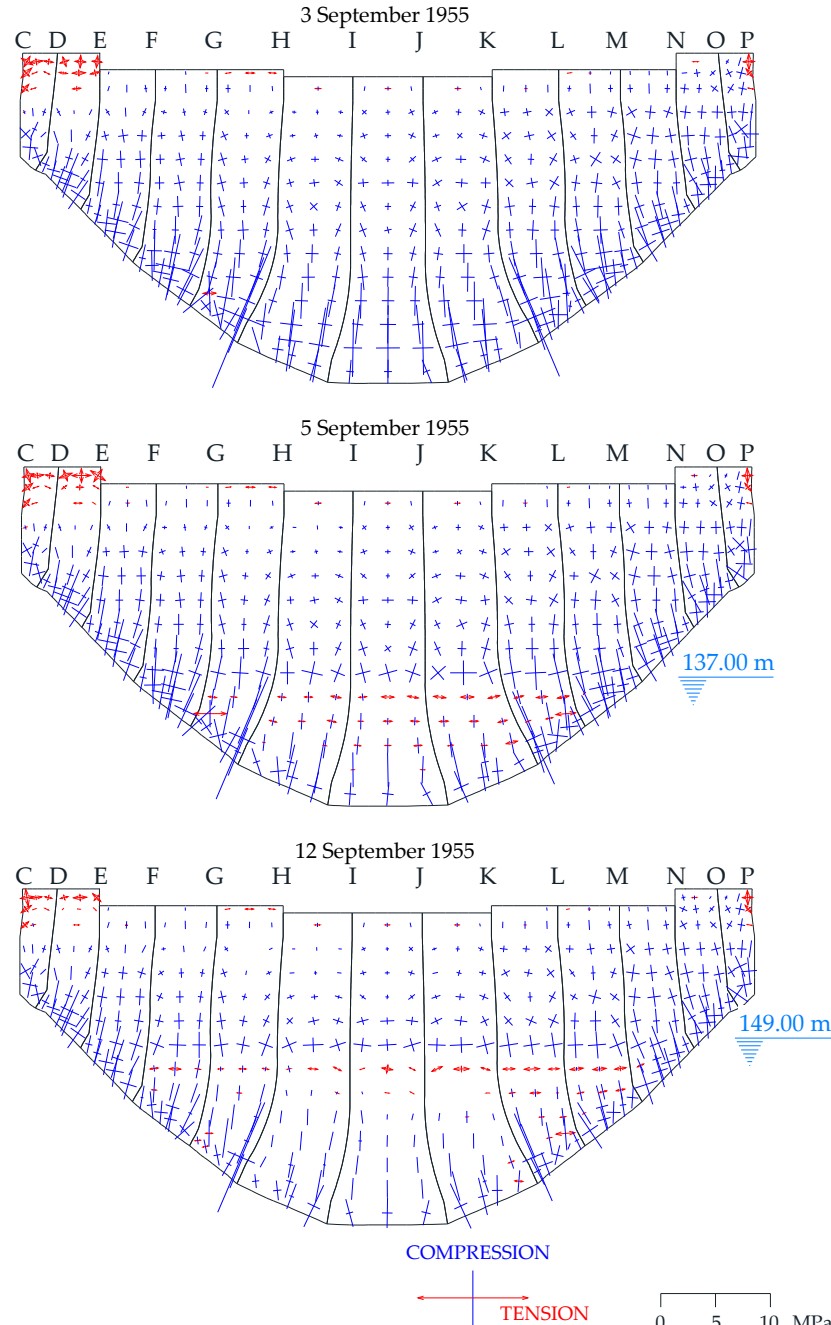

**Figure 14.** Distribution of principal stresses over the upstream surface for the empty reservoir with a water level of 137 m and 149 m.

As the water rose, the surface in contact with it cooled rapidly. The surface contraction due to the cooling was restrained by the hotter interior concrete, which did not contract as rapidly as the surface, and by the concrete located above the water level. As a result, an equilibrium deformation was obtained, inducing thermal compressive stresses in the hotter concrete and tensile stresses in the colder concrete, as can be observed in the principal stress distributions represented in the figures.

Up to the water level of 149 m, the self-weight counterbalanced the vertical (cantilever) stresses generated by the temperature variation. However, when the water level reached a level of 153 m, the thermal vertical stresses started to be predominant, with a maximum expression at a water level of 155 m. The nine days between the water levels of 155 m and

160 m were enough time to cool the emerged concrete and lower the vertical tensile stresses.

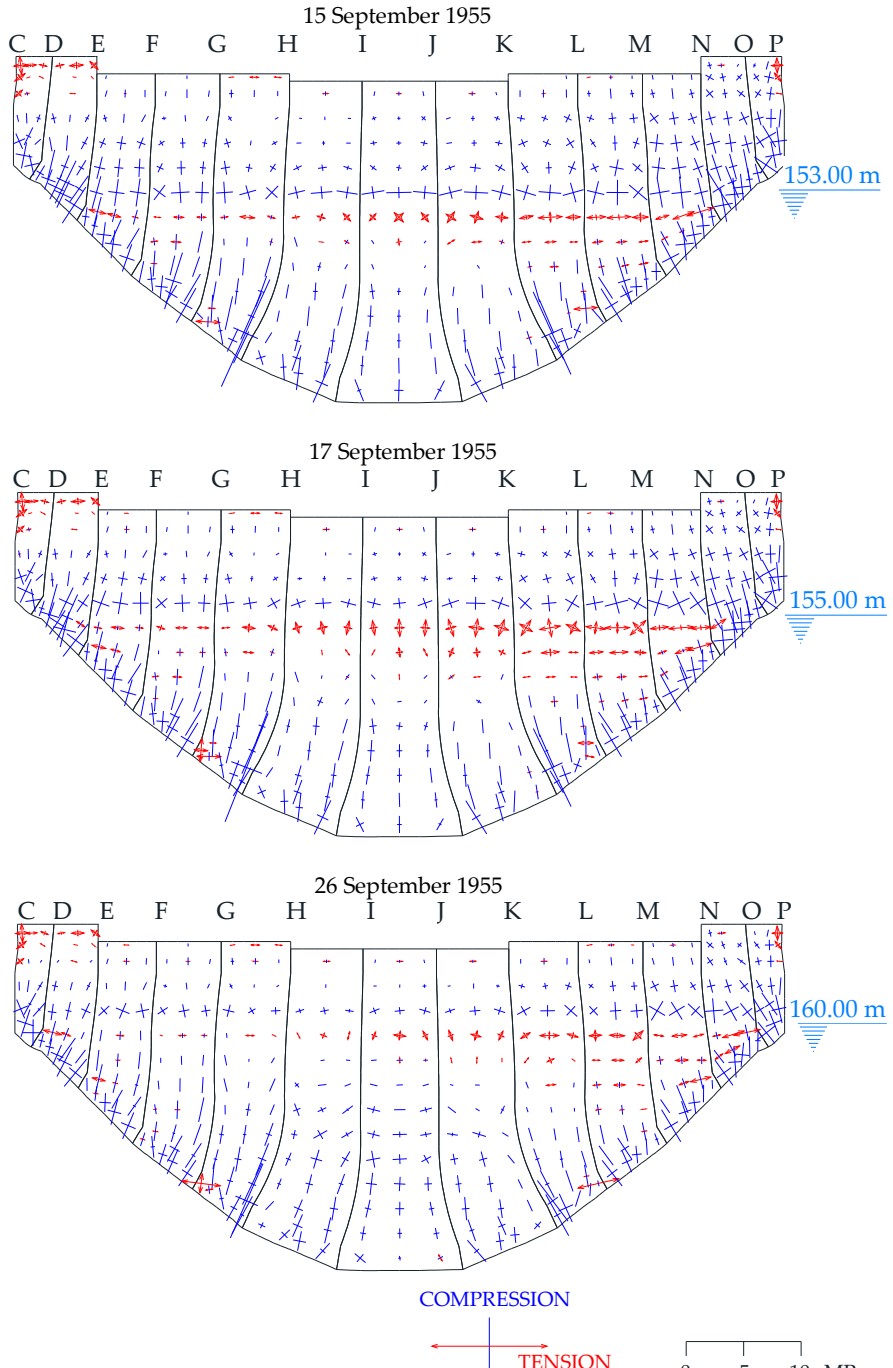

**Figure 15.** Distribution of principal stresses over the upstream surface for water levels of 153 m, 155 m and 160 m.

## 8. Prediction of Future Behavior

In order to corroborate that the presence of cracks has no influence on the static behavior of the dam, a comparison of the radial displacements measured by geodetic triangulation and those obtained with the finite element model are represented in Figure 16 for the period of 2001–2016.

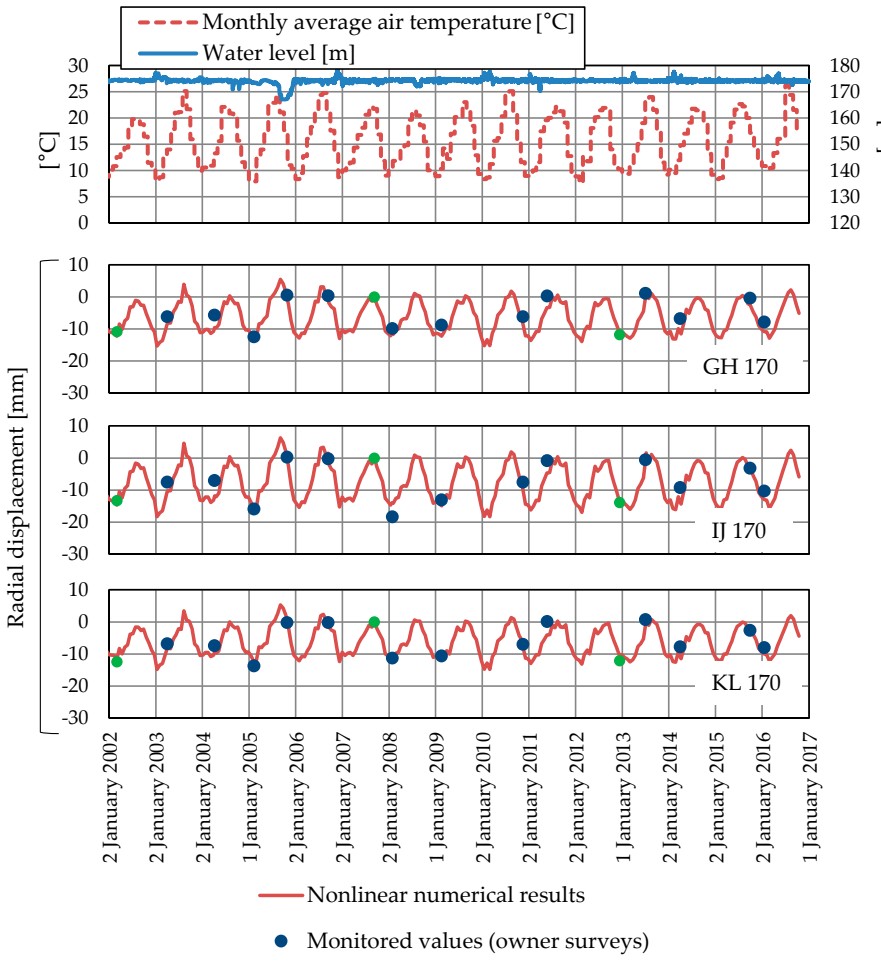

**Figure 16.** Monthly average air temperature, reservoir levels and monitored radial displacement between 2002 and 2016.

As for its potential seismic performance, an explanation can be found in the U.S. Army Corps of Engineering Manual EM 1110-2-6053, illustrated in Figure 17:

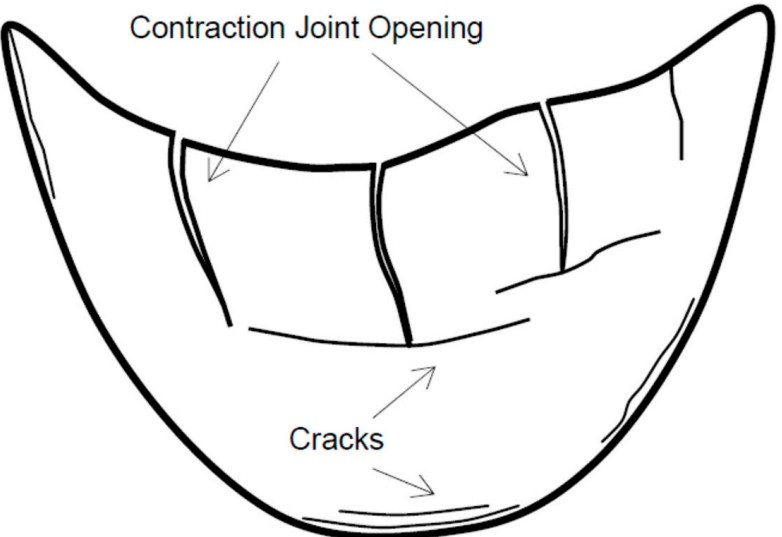

**Figure 17.** Response of arch dams to major earthquakes (adapted from [42]).

In arch dams, potentially opened contraction joints and cracked lift lines may subdivide the monolithic arch structure into partially free cantilever blocks, capable of transmitting only compressive or frictional forces. In this situation, any failure mode of the arch structure would more likely involve sliding stability of the partially free cantilevers. For small and moderate joint openings, the partially free cantilever blocks, bounded by opened joints, may remain stable through interlocking (wedging) with adjacent blocks. The extent of interlocking depends on the depth and type of shear keys and the amount of joint opening (EM 1110-2-6053 [42], p. 2–9).

## 9. Discussion and Conclusions

This paper presented the study, characterization and numerical simulation of the diagnosis procedure of the pathology affecting an old, thin dam. The cause of the main damage observed in the dam was attributed to the thermal cracking of weak construction joints during the initial impoundment. This hypothesis was validated through a chemo-thermo-mechanical finite element model.

For the thermal analysis of its construction, initial impoundment and first period of operation, a chemo-thermal model based on the chemical affinity concept was used.

Sequentially, a nonlinear viscoelastic analysis was performed in order to obtain the structural response of the dam. In this analysis, the contraction joints and the dam–foundation interface were modelled using zero-thickness interface elements under no-tension and no-sliding conditions, which avoided the need to determine new material parameters.

The conclusion of the analysis was that tensile vertical stresses developed on the upstream face of the dam during the initial impoundment together with weak construction joints caused the appearance of cracks in the upstream face of the dam. Over time, water penetration expanded these cracks, reaching the downstream face.

**Author Contributions:** Conceptualization, N.S.L.; methodology, N.S.L.; software, N.S.L. and E.C.; validation, N.S.L. and E.C.; formal analysis, N.S.L.; investigation, N.S.L. and E.C.; writing—original draft preparation, N.S.L.; writing—review and editing, N.S.L.; visualization, N.S.L.; supervision, N.S.L. All authors have read and agreed to the published version of the manuscript.

**Funding:** This research received no external funding.

**Data Availability Statement:** The data presented in this study are available in the article.

**Conflicts of Interest:** The authors declare no conflict of interest.

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
