# Peer review of "Chemo-Thermo-Mechanical FEA as a Support Tool for Damage Diagnostic of a Cracked Concrete Arch Dam: A Case Study"

_2673-4117, doi:10.3390/eng4020074_

Round 1
Reviewer 1 Report
The manuscript by Leitao and Castilho provides sufficient background and introduction of the problem. However, it lacks the mathematical rigor and the governing equations are not described adequately. Works on Thermal cracking already exists in literature, therefore improvements that have been made over existing models should be highlighted clearly.
Ideally, before section 6 it is recommended to have a section which mathematically describes the problem statement. Finite element method mentioned in section 6 is a numerical scheme to achieve solutions to the mathematical equation. Therefore to improve the clarity the authors to clearly state the mathematical equations they are solving along the boundary conditions.
Reviewer 2 Report
The authors describe the construction joints of an ancient concrete arch dam showing signs of reservoir seepage, resulting in calcium carbonate accumulation on the downstream face.Based on the analysis of the existing data, the hypothesis that the temperature gradient in the initial stage of dam construction promotes the opening of the upstream construction joints is put forward.The chemical-thermal-mechanical finite element analysis is used to simulate the behavior of the dam during construction and initial impoundment to verify the correctness of the proposed assumptions.However, if there are still some deficiencies in the published article, the author is requested to make supplementary explanations on the following issues:
1. Some figures in Figure 1 are not clearly displayed, and the author is advised to make adjustments.
2. It is observed in Fig. 8 that the mesh size of finite element structure calculation is large, and the larger mesh size may affect the accuracy of calculation results. It is suggested that the author supplement the influence of model mesh size on results.
3. Please check the number of the formula in the article. Some of them are wrong.
4. This paper describes the finite element calculation model of arch dam based on the concrete laying period of one week interval in the initial stage of arch dam construction, but it is not clear how this part is reflected in the model, and it is suggested that the author verify and supplement.
5. There are few descriptions about the loading of the finite element calculation model in the article, which are limited to the text of hydrostatic load, dead load and temperature load, and there is no specific data description. It is suggested that the author verify and supplement.
6. The author in the sixth section through the calculation of the radial displacement model and measured data value comparison confirmed that the cracks on the dam static analysis has no effect, but in fact the existence of some cracks in the dam body may lead to the local area of the dam stress concentration, the finite element calculation of the crack-free model may overestimate the safety of the dam, please talk about the relevant understanding.
7. It is suggested that the author should supplement the finite element model with cracks, analyze the local stress and strain distribution of arch dam model, and not only compare the measured radial displacement of dam body, and comprehensively evaluate the safety of arch dam structure.
Reviewer 3 Report
In this study, To illustrate the diagnostic process, an old concrete arch dam, which has shown signs of reservoir water seepage through some construction joints resulting in a buildup of calcium carbonate on the downstream face, is presented. After analyzing the available data, the hypothesis that the high temperature gradient promoted the opening of some construction joints on the upstream face during the first filling of the reservoir was put forward. Over time, water penetration expanded those cracks reaching the downstream face. To prove the diagnosis, a chemo-thermo-mechanical finite element analysis was carried out in order to simulate the behavior of the dam during its construction and the initial impoundment. In this regard, this study will be beneficial to the literature. It can be accepted after minor changes.
1. Language should be reviewed in general. Some places are difficult to understand by the reader.
2. Abstract is so long and irrelevant, especially in the beginning. It should be given information about the study.
3. In Fig.5, were these pictures taken by authors? If not, ıt should be given some references.
4. In section 6.1, for better reader understanding, authors should give the mesh properties. Because it was written that “The dam model comprises four layers of 20 node solid elements through its thickness. For practical reasons, the same solid elements mesh was used for both the thermal and the mechanical analysis”. What was the reason behind this?
5. Conclusion is so short. It should be given more specific results and discussions.
6. References should be updated. Especially authors should add the extra new researches such as
10.1007/s13349-023-00669-6 , 10.1080/13873954.2022.2033274,10.12989/gae.2021.24.5.443, 10.3390/civileng4010010
Round 2
Reviewer 1 Report
The comments are clearly addressed by the authors. The authors have also provided the governing equations. Providing the mathematical statement makes the paper much more clearer of what is being solved. Also, I would like to appreciate the authors for pointing out that the manuscript describes a diagnostic process rather than a new model.